# 0.3-Volt Rail-to-Rail DDTA and Its Application in a Universal Filter and Quadrature Oscillator

**DOI:** 10.3390/s22072655

**Published:** 2022-03-30

**Authors:** Fabian Khateb, Montree Kumngern, Tomasz Kulej, Dalibor Biolek

**Affiliations:** 1Department of Microelectronics, Brno University of Technology, Technická 10, 601 90 Brno, Czech Republic; khateb@vutbr.cz; 2Faculty of Biomedical Engineering, Czech Technical University in Prague, Nám. Sítná 3105, 166 36 Kladno, Czech Republic; 3Department of Telecommunications Engineering, School of Engineering, King Mongkut’s Institute of Technology Ladkrabang, Bangkok 10520, Thailand; 4Department of Electrical Engineering, Czestochowa University of Technology, 42-201 Czestochowa, Poland; kulej@el.pcz.czest.pl; 5Department of Electrical Engineering, Brno University of Defence, Kounicova 65, 662 10 Brno, Czech Republic; dalibor.biolek@unob.cz

**Keywords:** universal filter, quadrature oscillator, differential difference transconductance amplifier, analog signal processing

## Abstract

This paper presents the extremely low-voltage supply of the CMOS structure of a differential difference transconductance amplifier (DDTA). With a 0.3-volt supply voltage, the circuit offers rail-to-rail operational capability. The circuit is designed for low-frequency biomedical and sensor applications, and it consumes 357.4 nW of power. Based on two DDTAs and two grounded capacitors, a voltage-mode universal filter and quadrature oscillator are presented as applications. The universal filter possesses high-input impedance and electronic tuning ability of the natural frequency in the range of tens up to hundreds of Hz. The total harmonic distortion (THD) for the band-pass filter was 0.5% for 100 mV_pp_ @ 84.47 Hz input voltage. The slight modification of the filter yields a quadrature oscillator. The condition and the frequency of oscillation are orthogonally controllable. The frequency of oscillation can also be controlled electronically. The THD for a 67 Hz oscillation frequency was around 1.2%. The circuit is designed and simulated in a Cadence environment using 130 nm CMOS technology from United Microelectronics Corporation (UMC). The simulation results confirm the performance of the designed circuits.

## 1. Introduction

In recent years, extremely low-voltage operation capability and low-power consumption became inevitable requirements in modern, battery-operated, portable electronics and self-powered systems. In modern nanoscale complementary metal–oxide–semiconductor (CMOS) technologies, scaling the power supply voltage sustains the reliability and performance improvement of digital circuits; however, it causes performance degradation in the analog part. This poses a continual challenge for analog circuit designers to maintain acceptable performance for applications and systems-on-chip. The main impact of reducing the voltage supply on analog circuit performance, such as an operational amplifier (Op-Amp) or transconductance amplifier (TA or OTA), is the reduced input voltage swing, the transconductance value, and the voltage gain. A conventional design technique used to increase the input voltage swing is rail-to-rail circuits composed of both PMOS and NMOS differential pairs. However, these circuits are complex due to the additional differential pair, current branches, and circuitry used to maintain constant transconductance over the whole input voltage range. Therefore, non-conventional techniques, such as bulk-driven (BD) [1,2,3,4,5,6,7,8,9,10,11,12,13,14], floating-gate (FG), and quasi-floating-gate (QFG) [15,16], are suitable candidates for circuits operating with low supply voltages. They may reduce the threshold voltage or even remove it from the signal path, resulting in an extended input voltage range. Multiple-input MOS transistor (MI-MOST) is an alternative technique to the FG. However, unlike the FG, the MI-MOST: (a) does not need two polysilicon technologies; hence, it can be implemented in any standard CMOS technology; (b) it can process both AC and DC signals; and (c) there is no gate floating, and hence no issue associated with removing the initial charge trapped as in the case of FG. The multiple-input can be applied to the gate, to the bulk, or to their combination [17,18,19,20,21,22,23,24,25,26,27]. From the realization point of view, analog filter applications with MI-MOST may reduce the count of needed active devices [17,18,19,20,21,22,23,24,28,29]. This leads to simplified filter circuitry and reduced power consumption and chip area.

The universal filter and oscillator are important blocks for analog signal processing. Their applications include communication, control, and instrumentation systems [30,31,32]. Biquadratic filters and oscillators can be applied to biomedical systems [33,34,35]. Therefore, low-voltage supply and low-power consumption are mainly considered for these applications.

The differential difference transconductance amplifier (DDTA) is a useful analog block for filter applications [36,37,38,39,40]. It combines the features of a differential difference amplifier (DDA) with unity gain, like addition and subtraction voltage ability, high-input impedance, a low number of components, and the advantages of a operational transconductance amplifier (OTA), such as electronic tuning ability and simple circuitry. There are DDTA-based universal filters and oscillators available in the literature [36,37,38,39,40]. However, these DDTAs are not suitable for extremely low-voltage supply (i.e., ≤0.3 V) applications. Their structures are standard; hence, reducing their voltage supply leads to significant performance degradation, for instance a reduced input voltage swing. Focusing on recently published universal filters and/or oscillators [41,42,43,44,45,46,47], only the circuit in [48] can work with sub-volt supply (±0.3 V) and low-power consumption (5.77 µW).

Therefore, this paper presents an innovative CMOS structure for DDTA capable of working under a 0.3 V supply voltage with a rail-to-rail input voltage swing without degrading the other circuit’s performance. As an application of DDTA, a multiple-input, multiple-output (MIMO) universal filter is presented. The filter employs two DDTAs and two grounded capacitors. A variety of filter responses can be obtained by suitably applying the input signal and suitably choosing the output terminal. The natural frequency of filter responses can be electronically controlled. The proposed universal filter has also been modified to work as a quadrature oscillator. The frequency of oscillation can be controlled electronically. The proposed universal filter and quadrature oscillator can be applied to biomedical and sensor systems due to their extremely low voltage supply and low power consumption.

This paper is organized as follows: In Section 2, the DDTA and its innovative CMOS structure are presented; Section 3 presents its application in the voltage-mode universal filter and the quadrature oscillator; Section 4 presents the simulation results; and Section 5 concludes the paper.

## 2. DDTA and Its CMOS Structure

The symbol of the DDTA is shown in Figure 1. In the ideal case, this active component is described by the following equations:(1)$$\begin{array}{c} V_{w}=V_{y1}-V_{y2}+V_{y3}\\ I_{o}=g_{m}\left(V_{w}-V_{y4}\right) \end{array}\}$$

The CMOS structure of the proposed DDTA is shown in Figure 2. The circuit consists of two main blocks, namely, the differential-difference amplifier operating in a unity feedback configuration, thus forming a differential-difference current conveyor (DDCC), and the transconductance amplifier (TA). Both circuits are based on non-tailed differential amplifiers [1], which allow for operation in an ultra-low-voltage environment with rail-to-rail input swing.

The DDCC block consists of two stages, the input differential amplifier, M_1_–M_6_, and the class-A output stage, M_9_–M_10_. The capacitance *C_C_* is used for frequency compensation. Its value can be calculated in the same way as that for a two-stage operational amplifier. The input stage of the DDCC circuit can be seen as a non-tailed differential pair with an additional partial positive feedback (PPF) circuit. The solution, first presented in [2] and experimentally validated in [3], has been adopted here. The transistors, M_7_ and M_8_, generate negative conductances, -*g_m_*_7_ and -*g_m_*_8_, which partially compensate for the positive conductances of the diode-connected transistors, M_2A,B_ (≈*g_m_*_2_), thus increasing the resistances at the gate-drain nodes of these transistors, and consequently the voltage gain from inputs to the gate terminals of M_1A,B_. This improves the overall transconductance and voltage gain of the first stage.

In the proposed realization, the input transistors M_1A,B_ have been replaced by bulk-driven MI-MOST transistors. The symbol and CMOS realization of these devices are shown in Figure 3. This approach allows design simplification and the decreasing of the total dissipation power by removing one differential stage of the conventional DDCC. This is the result of the fact that summation of input signals is realized using the passive voltage divider/summing circuit composed of the capacitances *C_Bi_* (see Figure 3b). The capacitances are shunted by large resistances, R_MOSi_, that allow proper DC biasing of the bulk terminals of M_1A,B_. The resistors are realized as the antiparallel connection of two MOS transistors operating in a cutoff region, as shown in Figure 3c.

The low-frequency open-loop voltage gain of the DDCC, from one differential input, with the second input grounded for AC signals, can be expressed as follows:(2)$$A_{vo}=G_{m}\left(r_{ds1}||r_{ds6}\right)g_{m10}\left(r_{ds9}||r_{ds10}\right)$$
where *G_m_* is the transconductance of the input differential stage given by:(3)$$G_{m}≅\;\beta \frac{2g_{mb1}}{\left(1-m\right)}$$
where *β* is the voltage gain of the input capacitive divider, equal to ½ if all capacitances *C_Bi_* are equal to each other and the input capacitance of the MOS transistor from its bulk terminal can be neglected. The factor *m* represents the absolute value of the ratio of negative to positive conductances at the gate/drain nodes of M_2A,B_:(4)$$m=\frac{g_{m7,8}}{g_{m2}+g_{ds2}+g_{ds3}+g_{ds7,8}}≅\frac{g_{m7,8}}{g_{m2}}$$

Note that the transconductance *G_m_* as well as the voltage gain A_vo_ tend to infinity, as *m* tends to unity, namely, when the negative conductances generated by M_7_ and M_8_ fully compensate the positive conductances of M_2_, thus leading to infinite voltage gain from inputs to the drain/gate nodes of M_2A,B_. However, when the difference between *g_m_*_2_ and *g_m_*_7,8_ is decreasing, namely when *m* is increasing to unity, then the circuit sensitivity to transistor mismatch is increasing as well, which limits the maximum value of *m*. The second limitation is associated with the location of the parasitic pole associated with the PPF circuit, which is given by the formula
(5)$$\omega_{p}≅\frac{g_{m2}\left(1-m\right)}{C_{\Sigma }}$$
where *C*_∑_ is the total capacitance associated with the gate/drain nodes of M_2A,B_. Note that the frequency of this pole decreases with increasing *m*, namely, as the total resistance at the gate/drain nodes of M_2_ increases with increasing positive feedback. For stable operation, the pole should be located well above the GBW product of the internal DDA, which is
(6)$$\omega_{GBW}=\frac{G_{m}}{C_{C}}$$

In view of the above considerations, the output signal at the *W* terminal for low frequencies can be expressed as
(7)$$V_{w}=\frac{A_{vo}}{1+A_{vo}}\left(V_{y1}-V_{y2}+V_{y3}\right)$$

Note that accuracy of this function is improved thanks to the impact of the PPF, which enlarges the low-frequency voltage gain *A_vo_*. The 3 dB frequency of this function is approximately equal to *ω_GBW_*. The low-frequency output resistance at the *W* terminal is given as follows:(8)$$r_{outW}=\frac{r_{ds9}||r_{ds10}}{1+A_{vo}}$$

Thus, the resistance *r_outW_* is also improved (decreased) thanks to the larger value of *A_vo_*.

The second block of the proposed DDTA is the linear transconductance amplifier, TA. The circuit applied here was first proposed and validated experimentally in [4]. It can be considered as a non-tailed BD pair [1], linearized with an additional linear resistance R, which significantly improves the linearity of the circuit. Thanks to its non-tailed architecture, the circuit can operate from a very low-voltage supply.

Assuming that transistor M_B_ is identical with M_3_ and M_4_, the DC transfer characteristic of the TA in Figure 2 can be described by the formula [4]
(9)$$I_{O}=2I_{set}\left[\sinh\left(x\right)-\left(\mathrm{Ax}\right)\cosh\left(x\right)\right]$$
where
(10)$$A=\frac{n_{p}U_{T}}{I_{set}\left(R+\frac{2}{g_{m1,2}}\right)}$$
(11)$$x=\frac{\eta (V_{w}-V_{y4})}{n_{p}U_{T}}$$
and *n_p_* is the subthreshold slope factor for a p-channel MOS, *U_T_* is the thermal potential, and *η* = (*n_p_* − 1) = *g_mb_*_1,2_/*g_m_*_1,2_ is the bulk-to-gate transconductance ratio for transistors M_1_ and M_2_.

As it was shown in [4], if the following condition holds
(12)$$R=\frac{1}{g_{m1,2}}$$
then the circuit exhibits an optimum linearity. However, even for the non-optimal case, the linearity of Equation (9) is much better than for the original circuit without the resistance, *R*; therefore, the TA can be tuned using the current source *I_set_*, while still maintaining good linearity of its transfer characteristic.

The small-signal transconductance *g_m_* of the TA in the general case is
(13)$$g_{m}≅2g_{mb1,2}\left[\frac{R+\frac{1}{g_{m1,2}}}{R+\frac{2}{g_{m1,2}}}\right]$$
thus, in the optimum case, (*R* = 1/*g_m_*_1,2_), it is equal to 4*g_mb_*_1,2_/3.

## 3. Proposed Applications

### 3.1. Proposed Universal Filter

Figure 4 shows the proposed voltage-mode MIMO universal filter. The topology employs two DDTAs and two grounded capacitors. The terminals *V_in_*_1_, *V_in_*_2_, *V_in_*_3_, *V_in_*_4_, and *V_in_*_5_ provide high-input impedances, and the terminals *V_o_*_1_ and *V_o_*_3_ low-output impedances, whereas the terminals *V_o_*_2_ and *V_o_*_4_ require external buffer circuits if a low-impedance load is applied.

Using (1) and nodal analysis, the output voltages of Figure 4 can be expressed as follows:(14)$$V_{o1}=\frac{\left(s^{2}C_{1}C_{2}+sC_{2}g_{m1}\right)\left(V_{in1}+V_{in2}\right)+sC_{1}g_{m2}\left(V_{in3}-V_{in4}\right)+\left(sC_{1}g_{m2}+g_{m1}g_{m2}\right)V_{in5}}{s^{2}C_{1}C_{2}+sC_{2}g_{m1}+g_{m1}g_{m2}}$$
(15)$$V_{o2}=\frac{sC_{2}g_{m1}\left(V_{in1}+V_{in2}\right)+\left(sC_{2}g_{m1}+g_{m1}g_{m2}\right)\left(V_{in3}-V_{in4}\right)-g_{m1}g_{m2}V_{in5}}{s^{2}C_{1}C_{2}+sC_{2}g_{m1}+g_{m1}g_{m2}}$$
(16)$$V_{o3}=\frac{sC_{2}g_{m1}\left(V_{in1}+V_{in2}\right)+s^{2}C_{1}C_{2}\left(V_{in4}-V_{in3}\right)+g_{m1}g_{m2}V_{in5}}{s^{2}C_{1}C_{2}+sC_{2}g_{m1}+g_{m1}g_{m2}}$$
(17)$$V_{o4}=\frac{g_{m1}g_{m2}\left(V_{in1}+V_{in2}\right)+sC_{1}g_{m2}\left(V_{in4}-V_{in3}\right)-\left(sC_{1}g_{m2}+g_{m1}g_{m2}\right)V_{in5}}{s^{2}C_{1}C_{2}+sC_{2}g_{m1}+g_{m1}g_{m2}}$$

From (14)–(17), the low-pass (LP), band-pass (BP), high-pass (HP), band-stop (BS), and all-pass (AP) responses can be obtained by properly applying the input signal and choosing the output terminals as shown in Table 1. The input terminals that are not used should be connected to ground. In the case of the all-pass filtering response, the circuit requires an inverting-type input signal, which can be obtained using additional DDTA.

The natural frequency ($\omega_{o}$) and the quality factor (Q) of the filter can be respectively expressed as
(18)$$\omega_{o}=\sqrt{\frac{g_{m1}g_{m2}}{C_{1}C_{2}}}$$
(19)$$Q=\sqrt{\frac{g_{m2}C_{1}}{g_{m1}C_{2}}}$$

From (18) and (19), the natural frequency and the quality factor can be designed, as the quality factor can be given by $C_{1}$/$C_{2}$ by letting $g_{m1}\;$=$\;g_{m2}$ whereas the natural frequency can be obtained electronically by adjusting $g_{m}$ ($g_{m}\;$=$\;g_{m1}$ = $g_{m2}$).

### 3.2. Proposed Quadrature Oscillator

The proposed universal filter in Figure 4 was modified to work as a quadrature oscillator as shown in Figure 5. It can be obtained by using a non-inverting BP filtering response and a feedback connection. Using (14), the transfer function between $V_{o1}$ and $V_{in3}$ can be expressed as follows:(20)$$\frac{V_{o1}}{V_{in3}}=\frac{sC_{1}g_{m2}}{s^{2}C_{1}C_{2}+sC_{2}g_{m1}+g_{m1}g_{m2}}$$

Letting $V_{o1}/V_{in3}$ = 1, the oscillator characteristic can be derived as
(21)$$s^{2}C_{1}C_{2}+s\left(C_{2}g_{m1}-C_{1}g_{m2}\right)+g_{m1}g_{m2}=0$$

Letting $g_{m1}=g_{m2}=g_{m}$, the condition of oscillation (CO) is
(22)$$C_{1}=C_{2}$$
and the frequency of oscillation (FO) is
(23)$$\omega_{o}=\sqrt{\frac{g_{m1}g_{m2}}{C_{1}C_{2}}}$$

Thus, the CO of the oscillator can be controlled by $C_{1}$ and/or $C_{2}$, and letting $g_{m1}\;$=$\;g_{m2}$, the FO can be controlled electronically by $g_{m}$ ($g_{m}\;$=$\;g_{m1}\;$=$\;g_{m2}$). Therefore, the FO and CO of the oscillator can be orthogonally controlled. The nodes *V_o_*_3_ and *V_o_*_4_ provide quadrature output signals. It can be confirmed by the relationship between *V*_*o*3_ and *V*_*o*4_:(24)$$\frac{V_{o4}}{V_{o3}}=\frac{g_{m2}}{sC_{2}}$$

Thus, the phase difference between *V_o_*_3_ and *V_o_*_4_ is 90°. After setting *s* = *jω*_0_ into (24) and taking into account Equations (22) and (23) and the condition *g_m_*_1_ = *g_m_*_2_, the ratio (24) is one; thus, if oscillation condition (22) is accomplished, the oscillator provides equal amplitudes of both quadrature signals independently of the oscillation frequency.

### 3.3. Non-Idealities Analysis

Considering non-idealities of the DDTA, (1) can be rewritten as:(25)$$\begin{array}{c} V_{w}=\beta_{i1}V_{y1}-\beta_{i2}V_{y2}+\beta_{i3}V_{y3}\\ I_{o}=g_{mni}\left(V_{w}-V_{y4}\right) \end{array}\}$$
where $\beta_{i1}$ denotes the voltage gain from $V_{y1}$ to $V_{w}$ of i-th DDTA, $\beta_{i2}$ denotes the voltage gain from $V_{y2}$ to $V_{w}$ of i-th DDTA, and $\beta_{i3}$ denotes the voltage gain from $V_{y2}$ to $V_{w}$ of i-th DDTA. Ideally, the voltage gains $\beta_{i1}$, $\beta_{i2}$, and $\beta_{i3}$ are unity. The $g_{mni}$ is the non-ideal transconductance gain of the DDTA, whose frequency dependence is given by parasitic capacitance *C_o_* and resistance *R_o_* at o-terminal. In the frequency range near the cutoff frequency, $g_{mni}$ can be approximated as [48]
(26)$$g_{mni}\left(s\right)≅g_{mi}\left(1-\mu_{i}s\right)$$
where $\mu_{i}=1/\omega_{gmi}$, $\omega_{gi}$ denotes the first-order pole.

Using (25), the output voltages of Figure 4 can be rewritten to the form
(27)$$V_{o1}=\frac{\begin{array}{c} \left(s^{2}C_{1}C_{2}+sC_{2}g_{mn1}\beta_{21}\right)\left(\beta_{11}V_{in1}+\beta_{13}V_{in2}\right)+sC_{1}g_{mn2}\beta_{12}\left(\beta_{22}V_{in3}-\beta_{23}V_{in4}\right)\\ +\left(sC_{1}g_{mn2}\beta_{12}+g_{mn1}g_{mn2}\beta_{12}\beta_{21}\right)V_{in5} \end{array}}{s^{2}C_{1}C_{2}+sC_{2}g_{mn1}\beta_{21}+g_{mn1}g_{mn2}\beta_{12}\beta_{21}}$$
(28)$$V_{o2}=\frac{\begin{array}{c} sC_{2}g_{mn1}\left(\beta_{11}V_{in1}+\beta_{13}V_{in2}\right)+\left(sC_{2}g_{mn1}+g_{mn1}g_{mn2}\beta_{12}\right)\left(\beta_{22}V_{in3}-\beta_{23}V_{in4}\right)\\ -g_{mn1}g_{mn2}V_{in5} \end{array}}{s^{2}C_{1}C_{2}+sC_{2}g_{mn1}\beta_{21}+g_{mn1}g_{mn2}\beta_{12}\beta_{21}}$$
(29)$$V_{o3}=\frac{sC_{2}g_{mn1}\beta_{21}\left(\beta_{11}V_{in1}+\beta_{13}V_{in2}\right)+s^{2}C_{1}C_{2}\left(\beta_{23}V_{in4}-\beta_{22}V_{in3}\right)+g_{mn1}g_{mn2}\beta_{12}\beta_{21}V_{in5}}{s^{2}C_{1}C_{2}+sC_{2}g_{mn1}\beta_{21}+g_{mn1}g_{mn2}\beta_{12}\beta_{21}}$$
(30)$$V_{o4}=\frac{\begin{array}{c} g_{mn1}g_{mn2}\beta_{21}\left(\beta_{11}V_{in1}+\beta_{13}V_{in2}\right)+sC_{1}g_{mn2}\left(\beta_{23}V_{in4}-\beta_{22}V_{in3}\right)\\ -\left(sC_{1}g_{mn2}+g_{mn1}g_{mn2}\beta_{21}\right)V_{in5} \end{array}}{s^{2}C_{1}C_{2}+sC_{2}g_{mn1}\beta_{21}+g_{mn1}g_{mn2}\beta_{12}\beta_{21}}$$

Considering the denominator D(s) of (27)–(30), the modified parameters $\omega_{o}$ and Q can be expressed by:(31)$$\omega_{o}=\sqrt{\frac{g_{mn1}g_{mn2}\beta_{12}\beta_{21}}{C_{1}C_{2}}}$$
(32)$$Q=\sqrt{\frac{g_{mn2}C_{1}\beta_{12}}{g_{mn1}C_{2}\beta_{21}}}$$

From (27), the modified oscillator characteristic can be expressed as
(33)$$s^{2}C_{1}C_{2}+\left(sC_{2}g_{mn1}\beta_{21}-sC_{1}g_{mn2}\beta_{12}\beta_{22}\right)+g_{mn1}g_{mn2}\beta_{12}\beta_{21}=0$$

The modified CO and FO of the oscillator are then
(34)$$C_{1}\beta_{12}\beta_{22}=C_{2}\beta_{21}$$
(35)$$\omega_{o}=\sqrt{\frac{g_{mn1}g_{mn2}\beta_{12}\beta_{21}}{C_{1}C_{2}}}$$

Since this work is focused on circuits that operate at low frequency, Equation (26) is not taken in consideration. In the case that the universal filter and the quadrature oscillator operate in the frequency range in which the frequency dependence of *g*_m_ asserts its influence, then (26) should be used to refine the error analysis.

## 4. Simulation Results

The DDTA circuit and its applications were designed in a Cadence environment, using 130 nm CMOS technology from UMC. The transistor’s aspect ratio and values of passive devices are included in Table 2. The voltage supply is 0.3 V (V_DD_ = −V_SS_ = 0.15 V), the bias current of the DDCC I_B_ = 50 nA, and the nominal value of the setting current of the TA *I_set_* = 500 nA. The nominal power consumption of the DDTA is 357.4 nW (DDCC = 70.21 nW, TA = 287.2 nW). The input and compensation capacitors are highly linear metal–isolator–metal capacitors (MIM). The linear resistor *R* is a high-resistance poly-resistor.

The open-loop gain of the DDCC (i.e., without the unity gain feedback) was simulated as 73.9 dB, and the phase margin was 56.2° for 20 pF load capacitor. The simulated magnitude characteristics of the DDCC are shown in Figure 6. The low-frequency gain for V_W_/V_Y1_ (=V_W_/V_Y3_) and V_W_/V_Y2_ is 14 mdB and 57.29 mdB, while the −3 dB bandwidth is 22.24 kHz and 22.23 kHz, respectively.

The simulated DC transfer characteristics of the DDCC are shown in Figure 7. As is evident, the DDCC enjoys rail-to-rail operation for all its inverting and non-inverting inputs. This rail-to-rail operation capability is a design achievement.

The simulated gain and phase characteristics for the TA with *I_set_* = 0.5 µA and 20 pF load capacitance are shown in Figure 8. The low DC gain is 23.2 dB, and the bandwidth (BW) is 19.65 kHz, while the phase error is 3.8°.

Figure 9a,b shows the DC characteristic of the output current and the transconductance versus fully differential input voltage V_in_ (V_in_ = V_+_ − V_y4_) for the TA for *I_set_* = 0.125 µA, 0.25 µA, and 0.5 µA. The rail-to-rail operation with high linearity is evident.

To determine the impact of mismatch and process variation on the circuit’s performance, Monte Carlo (MC) simulations (200 runs) were carried out. As the histograms show in Figure 10, the impact of mismatch and process variation on the gain and −3 dB BW of the DDCC is low. The impact is also low on the gain and phase error of the TA as shown in Figure 11.

The process, voltage, temperature (PVT) corners analysis was also provided on the proposed DDTA. The MOS transistor corners were slow-slow, slow-fast, fast-slow, and fast-fast, the voltage supply corners were (V_DD_ − V_SS_) ± 10%, and the temperature corners were −20 °C and 70 °C. Table 3 and Table 4 show the results of the minimum, nominal, and maximum values of the gain, −3 dB BW for the DDCC, and gain and phase error for the TA. The impact of the PVT corners in all cases is acceptable.

The universal filter in Figure 4 was simulated for *C*_1_ = *C*_2_ = 5 nF, which are off-chip capacitors. The magnitude characteristics of the LPF, HPF, BPF, BSF, and APF are shown in Figure 12. The simulated natural frequency (*f*_o_) is around 81.47 Hz. It is worth mentioning that, due to the limited value of the output resistance of the TA, which is around 5.1 MΩ, the attenuations of the HPF and BPF are degraded at low frequencies. Therefore, if an application demands higher attenuation, then the output resistance could be increased, employing the MOS transistor self-cascode technique.

The BPF was tested by applying a sine wave signal *V_in_* = 100 mV_pp_ @ 81.47 Hz. The waveforms of the input and output voltages are shown in Figure 13a. The spectrum of the output signal is shown in Figure 13b, where the total harmonic distortion (THD) of the BPF output is 0.5%.

The electronic tuning capability of the LPF, BPF, HPF, and BSF with different bias currents, *I_set_* = 0.125, 0.25, 0.5, and 0.75 µA, is shown in Figure 14. The *f*_o_ was in the range of 21.11 Hz, 41.63 Hz, 81.47 Hz, and 115.74 Hz, respectively.

The simulation results showing the start of the oscillation and the steady state of the quadrature oscillator from Figure 5 are given in Figure 15. The oscillation frequency is 67 Hz, and the THD for outputs V_3_ and V_4_ are 1.2% and 1.29%, respectively.

Finally, Table 5 provides the comparison of the proposed filter with others in the literature [23,41,42,44,46,47]. It is evident that the proposed filter offers the highest number of filtering functions with lowest power supply and power consumption, thanks to the innovative CMOS structure of the DDTA.

## 5. Conclusions

This paper presents an innovative structure of a DDTA capable of operating under an extremely low voltage supply of 0.3 V while offering a rail-to-rail input voltage swing. As an application, a universal filter and quadrature oscillator based on two DDTAs and two grounded capacitors are presented. The simulation results including Monte Carlo and PVT analysis confirm the good functionality of the proposed circuits.

## Figures and Tables

**Figure 1 sensors-22-02655-f001:**
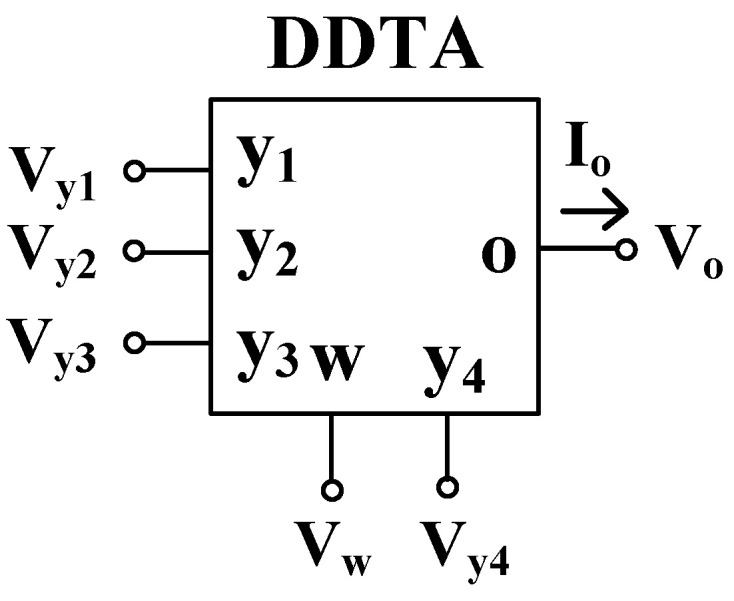
Symbol of DDTA.

**Figure 2 sensors-22-02655-f002:**
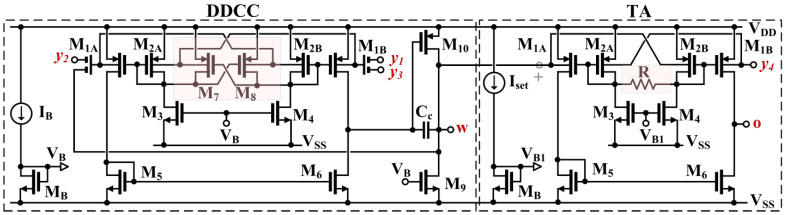
Proposed CMOS structure of DDTA.

**Figure 3 sensors-22-02655-f003:**
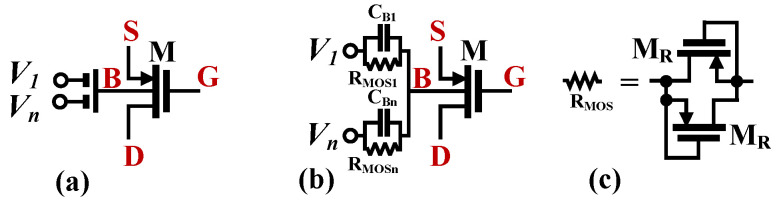
Bulk-driven MI-MOST: (**a**) symbol, (**b**) its realization, and (**c**) realization of RMOS.

**Figure 4 sensors-22-02655-f004:**
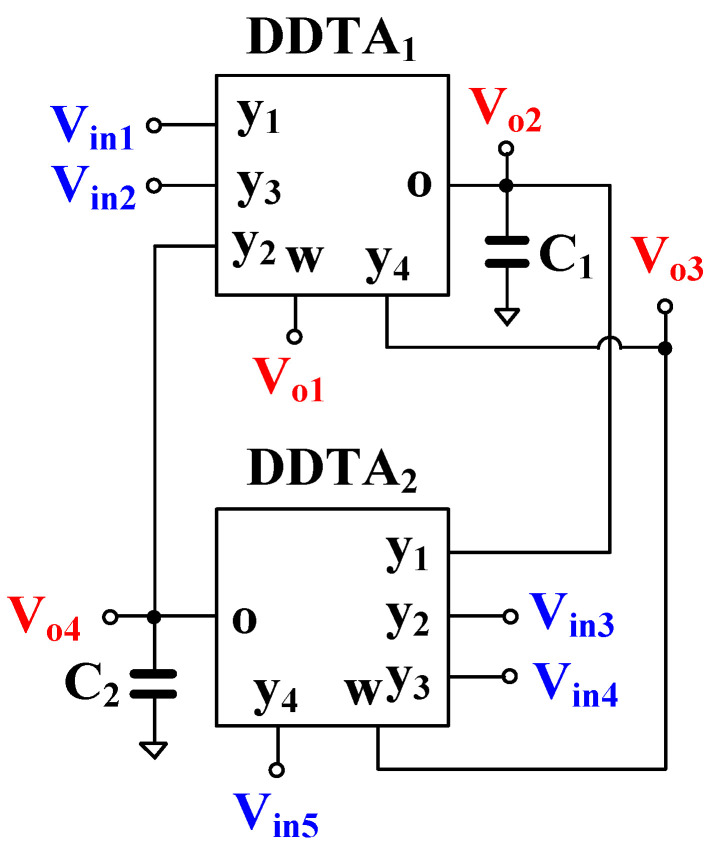
MIMO universal filter using DDTAs.

**Figure 5 sensors-22-02655-f005:**
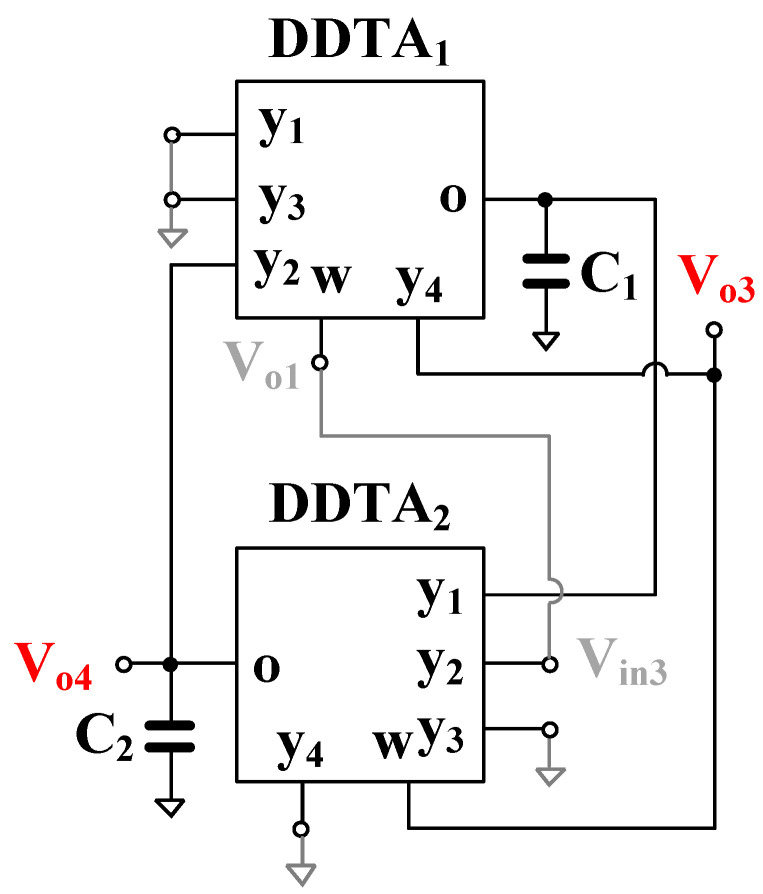
The quadrature oscillator.

**Figure 6 sensors-22-02655-f006:**
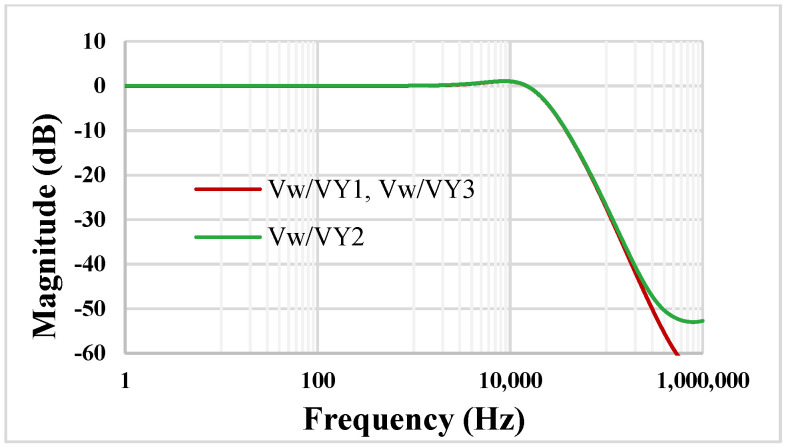
The magnitude characteristics of the DDCC.

**Figure 7 sensors-22-02655-f007:**
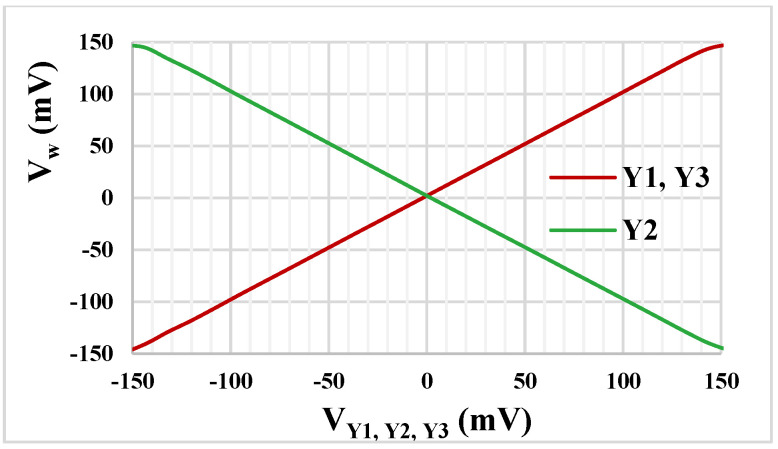
The DC transfer characteristics of the DDCC.

**Figure 8 sensors-22-02655-f008:**
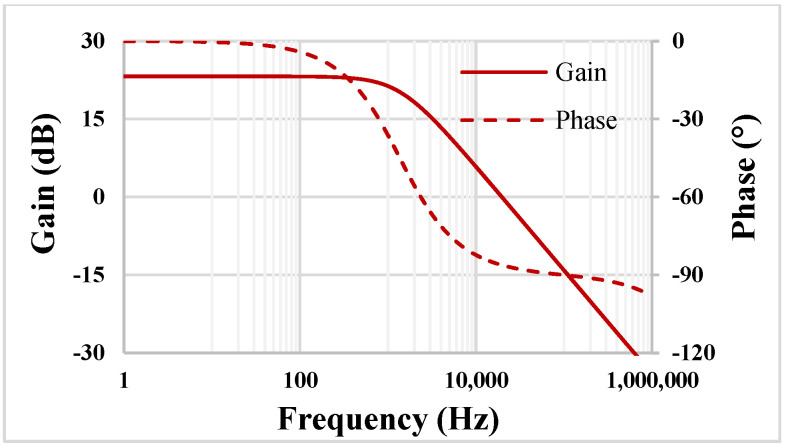
The gain and phase characteristics of the TA.

**Figure 9 sensors-22-02655-f009:**
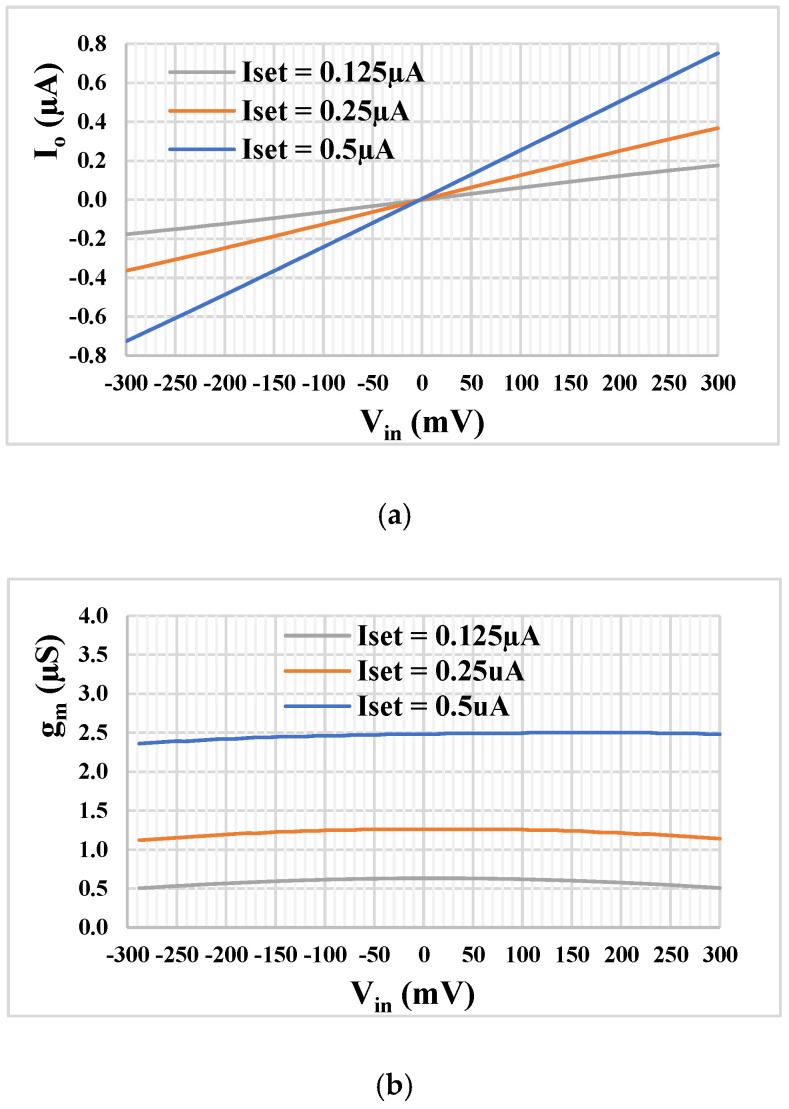
The DC output current (**a**) and transconductance *g_m_* (**b**) characteristics of the TA versus input voltage.

**Figure 10 sensors-22-02655-f010:**
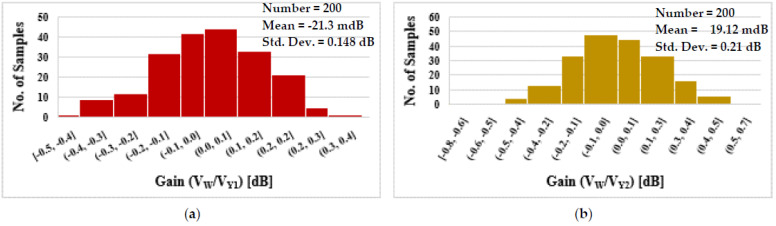
The DDCC histogram of the gain V_w_/V_Y1_ (**a**), V_w_/V_Y2_ (**b**), and the −3 dB BW V_w_/V_Y1_ (**c**), and V_w_/V_Y2_ (**d**).

**Figure 11 sensors-22-02655-f011:**
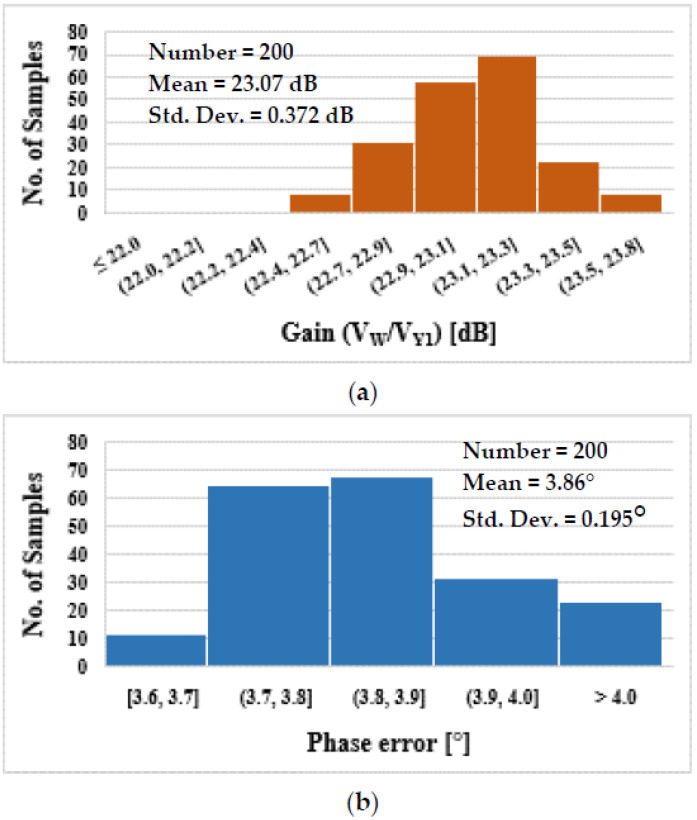
The TA histogram of the gain (**a**) and phase error (**b**).

**Figure 12 sensors-22-02655-f012:**
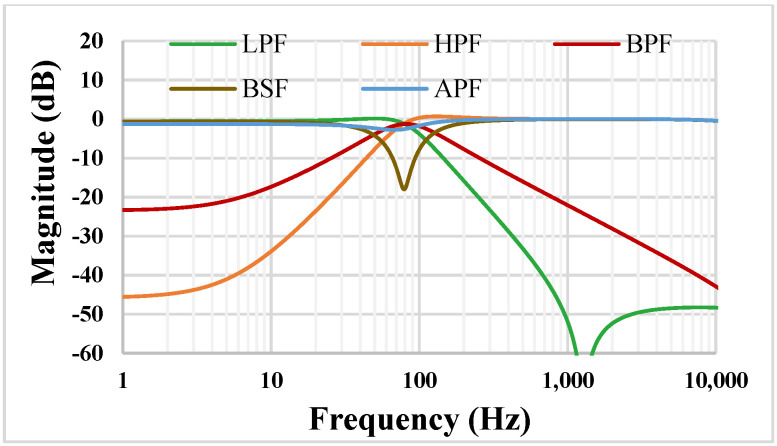
The magnitude characteristics of the universal filter.

**Figure 13 sensors-22-02655-f013:**
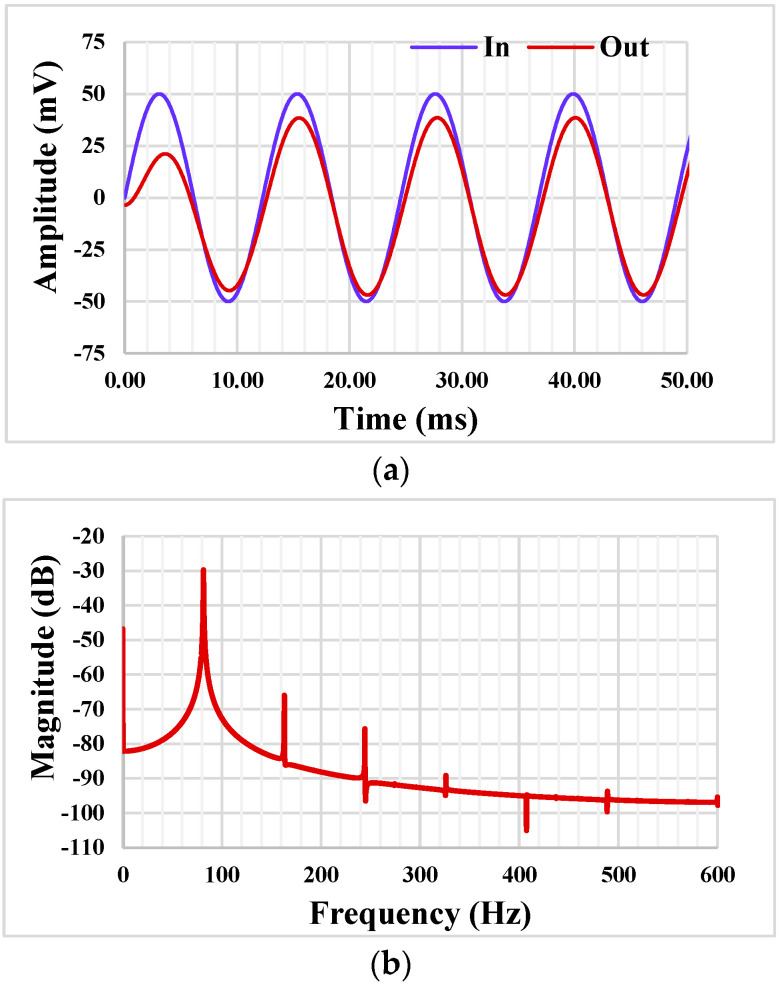
The transient response of the BPF (**a**) and the spectrum of the output signal (**b**).

**Figure 14 sensors-22-02655-f014:**
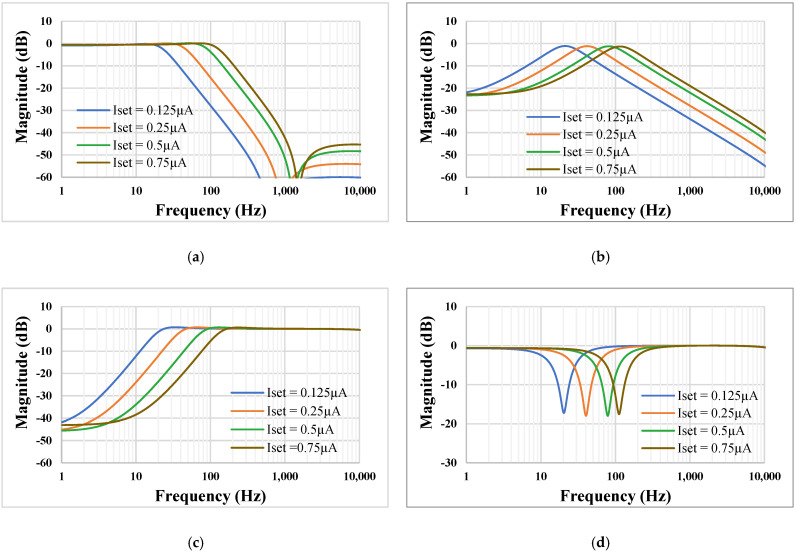
Magnitude characteristics showing the tuning capability of the LPF (**a**), BPF (**b**), HPF (**c**), and BSF (**d**) with different bias currents.

**Figure 15 sensors-22-02655-f015:**
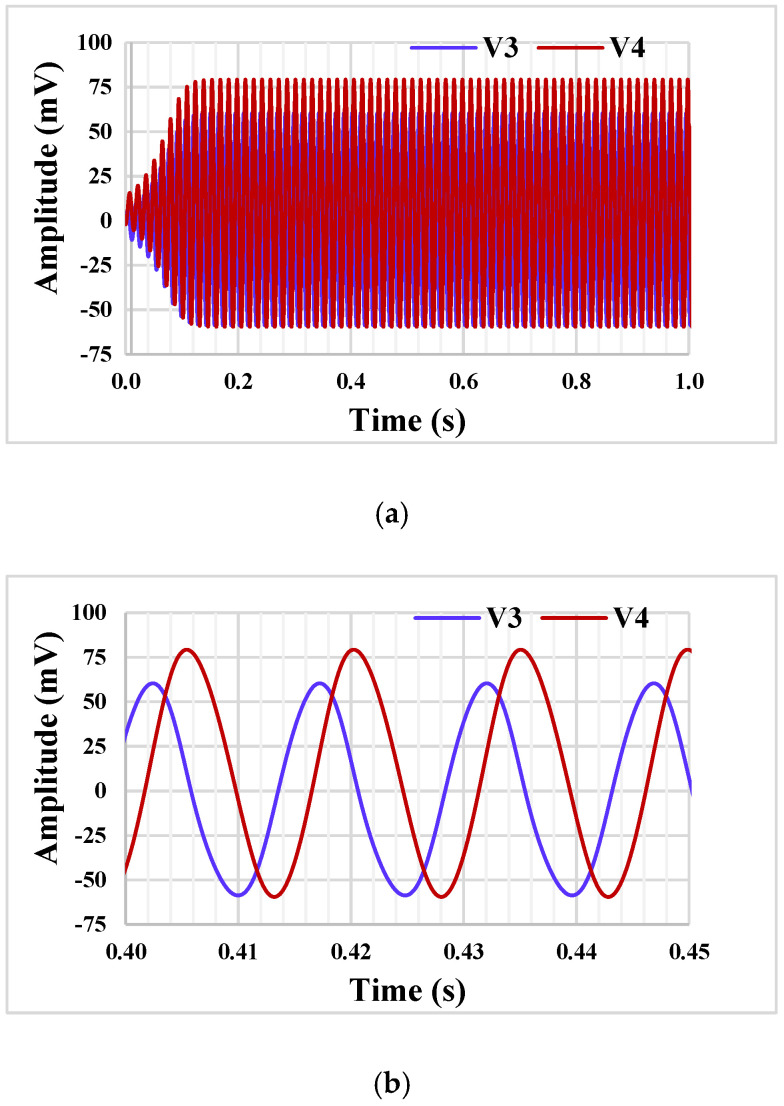
Starting the oscillation (**a**) and the steady state (**b**).

**Table 1 sensors-22-02655-t001:** Obtaining variant filtering functions of the proposed filter.

Filtering Function		Input	Output
LP	Non-inverting	$V_{in4}=V_{in5}$	$V_{o1}$
	Non-inverting	$V_{in5}$	$V_{o2}$
	Inverting	$V_{in1}=V_{in4}$	$V_{o2}$
	Non-inverting	$V_{in5}$	$V_{o3}$
	Non-inverting	$V_{in1}$	$V_{o4}$
	Non-inverting	$V_{in2}$	$V_{o4}$
	Inverting	$V_{in4}=V_{in5}$	$V_{o4}$
BP	Non-inverting	$V_{in3}$	$V_{o1}$
	Inverting	$V_{in4}$	$V_{o1}$
	Non-inverting	$V_{in1}$	$V_{o2}$
	Non-inverting	$V_{in2}$	$V_{o2}$
	Inverting	$V_{in4}=V_{in5}$	$V_{o2}$
	Non-inverting	$V_{in1}$	$V_{o3}$
	Non-inverting	$V_{in2}$	$V_{o3}$
	Non-inverting	$V_{in4}$	$V_{o4}$
	Inverting	$V_{in3}$	$V_{o4}$
	Inverting	$V_{in1}=V_{in5}$	$V_{o4}$
HP	Non-inverting	$V_{in1}=V_{in4}$	$V_{o1}$
	Inverting	$V_{in3}$	$V_{o3}$
	Non-inverting	$V_{in4}$	$V_{o3}$
BS	Non-inverting	$V_{in4}=V_{in5}$	$V_{o3}$
AP	Non-inverting	$-V_{in2}=V_{in4}=V_{in5}$	$V_{o3}$

**Table 2 sensors-22-02655-t002:** Transistor aspect ratios of the DDTA.

Device	W/L (µm/µm)
M_1A_, M_2A_, M_1B_, M_2B_	20/3
M_7_, M_8_	15/3
M_3_–M_6_, M_B_	10/3
M_9_	6 × 10/3
M_10_	6 × 20/3
M_R_	5/3
MIM capacitor: *C_B_* = 0.2 pF, *Cc* = 4 pF	
Poly-resistor *R* = 90 kΩ	

**Table 3 sensors-22-02655-t003:** The PVT corner analysis results for the DDCC.

DDCC	min.	nom.	max.
	P/V/T		P/V/T
Gain V_W_/V_Y1_ [mdB]	−75.3/9.8/−224	14	29.4/14/14
Gain V_W_/V_Y2_ [mdB]	−14.1/45.8/−75	57	101/67.3/57
−3 dB V_W_/V_Y1_ [kHz]	20.2/22/21	22.24	25.2/22.1/23.7
−3 dB V_W_/V_Y2_ [kHz]	20.1/22/20.8	22.23	25/22.7/23.4

**Table 4 sensors-22-02655-t004:** The PVT corner analysis results for the TA.

TA	min.	nom.	max.
	P/V/T		P/V/T
Gain [dB]	23/20.6/21.9	23.19	23.2/25.2/24
Phase error [°]	3.7/2.9/3.4	3.8	3.8/5.1/4.3
*G_m_* [µS]	2.2/2.2/2.2	2.48	2.5/2.5/2.4

**Table 5 sensors-22-02655-t005:** Comparison table with other universal filters.

Features	Proposed	Ref. [23]	Ref. [41]	Ref. [42]	Ref. [44]	Ref. [46]	Ref. [47]
Active and passive elements	2 DDTA, 2 C	5 OTA, 2 C	5 OTA, 2 C	4 OTA, 2 C	3 CFOA, 2 C, 4 R	3 VDBA, 2 C, 1 R (Figure 2)	8 OTA, 2 C (Figure 3)
Realization	CMOS structure (130 nm)	CMOS structure (180 nm) & commercial IC	commercial IC	commercial IC	CMOS structure (180 nm)	CMOS structure (180 nm)	CMOS structure (180 nm)
Filter type	MIMO	MISO	MIMO	MIMO	MOMO	MISO	MIMO
Number of filtering functions	22 (VM)	11 (VM)	13 (VM)	9 (VM)	5 (VM)	20 (Mixed-mode)	20 (Mixed-mode)
Offer universal filter and oscillator	Yes	Yes	Yes	Yes	Yes	No	No
Electronic control of parameter $\omega_{o}$	Yes	Yes	Yes	Yes	No	Yes	Yes
Natural frequency (kHz)	0.08147	1	217	144.7	757.88.	16.631 × 10^3^	5.77
Total harmonic distortion (%)	0.5@100 mV_pp_	1.67@600 mV_pp_	1.93@200 mV_pp_	3.83@170 mV_pp_	3.18@1.2 V_pp_	<3@500 mV_pp_	<2@200 mV_pp_
Power supply voltages (V)	0.3	1.2	±15	±15	±0.9	±1.25	±0.3
Simulated power consumption (µW)	0.715	96	860 × 10^3^	0.92 × 10^6^	5.4 × 10^3^	5.482 × 10^3^	5.77
Verification of result	Sim	Sim/Exp	Sim/Exp	Sim/Exp	Sim/Exp	Sim/Exp	Sim

Note: VDBA = voltage differencing buffered amplifier, VM = voltage-mode.

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
