# Peer review of "0.3-Volt Rail-to-Rail DDTA and Its Application in a Universal Filter and Quadrature Oscillator"

_sensors, 2022, doi:10.3390/s22072655_

Round 1

Reviewer 1 Report

The paper presents a so-called rail-to-rail DDTA used as a universal filter and other applications. The paper is well written and interesting. It presents a type of circuit that is apparently new (at least to the reviewer). The author should consider the following comments : 

  1. In the literature review, the author reports on DDTA, DDCC and MI-MOST. Most of the references provided (if not all) refer to papers from the same groups / authors. It would be interesting to provide additional references coming from author groups not related to the authors.
  2. The paper presents simulation results. Is it planned to fabricate the device? Where DDTA already fabricated (by other groups)? If so, the author should add the references to the literature review.
  3. Are the simulations presented post-layout simulation? Post-layout simulation should be provided. Figures of the layout should be provided.

Author Response

Reviewer 1

The paper presents a so-called rail-to-rail DDTA used as a universal filter and other applications. The paper is well written and interesting. It presents a type of circuit that is apparently new (at least to the reviewer). The author should consider the following comments : 

Comment #1. In the literature review, the author reports on DDTA, DDCC and MI-MOST. Most of the references provided (if not all) refer to papers from the same groups / authors. It would be interesting to provide additional references coming from author groups not related to the authors.

Response:

Thank you for the positive evaluation and comments. We have added more recent publications to the manuscript [44-48]. The references [44, 45] provide universal filter and quadrature oscillator in same manner as the proposed filter. The references [46, 47] provide many transfer functions (current-mode, voltage-mode, transadmittance-mode, transimpedance-mode filters) into single topology and the reference [48] is a universal filter that operate with sub-volte supply (±0.3V). The references [45, 47, 48] have been added in the comparison Table 4.

Comment # 2. The paper presents simulation results. Is it planned to fabricate the device? Where DDTA already fabricated (by other groups)? If so, the author should add the references to the literature review.

Response:

In the meantime, we are not planning to fabricate the DDTA. However, the 0.3V high linear rail-to-rail TA was fabricated and experimentally verified by us in [4].

To the best of authors knowledge, there is no paper presenting a fabricated DDTA. All previous DDTAs have a standard structure i.e. standard two differential pairs DDA followed by a standard OTA. These DDTAs are not innovative and not suitable for low-voltage low-power applications.

Reviewer 2 Report

Page 2, line 49. Not “polys silicon” but “polysilicon”.

Page 2, line 70. Not “the input, but “the reduced input.”

Author Response

Reviewer 2

Page 2, line 49. Not “polys silicon” but “polysilicon”.

Page 2, line 70. Not “the input, but “the reduced input.”

Response:

Thank you for the comments. We have correct the typos.

Reviewer 3 Report

  The manuscript prezents an ultra low power and rail-to-rail DDTA and two applications of the proposed DDTA: an universal filter and a quadrature oscillator. These applications are limited to low frequency operation; moreover, the universal biquad and the oscillator require large capacitors (2X 5nF), difficult to integrate.

 A few comments and suggestions for improving the manuscript are provided in the following.

 Abstract - should contain metrics of the performance parameters, tuning range, etc. not only power consumption

 line 177: "therefore the TA can be tuned using the current source Iset". Please provide details regarding the circuit implementation of current sources Iset and IB. Do these current souces have enough voltage headroom: VDD - VGS? What is the VGS of  dide-connected MB transistors?

 Language and arguments should be revised; some examples are given below:

 line 32 - became instead of become 
 Line 40 - remove "As"
 revise: "circuits operating under low voltage supply" - line 46
 revise: Line 55: "This leads to simplyfying [a simplified] filter circuitry and reducing [a reduced] power consumption..."
 line 57: replace applications with blocks/circuits

 Too general statements on lines 62, 63, 72, 290; replace "interesting analog block", "good features", "other performance acceptable" , "This is a good achievement of this design." with more precise statements.

 Line 78-80: please explain why the proposed circuits can be used in "biomedical and sensor systems"

 Info regarding CMOS technology from Table 4 contradicts the info from the abstract and Sim results: 180nm vs 130nm.

Author Response

  The manuscript prezents an ultra low power and rail-to-rail DDTA and two applications of the proposed DDTA: an universal filter and a quadrature oscillator. These applications are limited to low frequency operation; moreover, the universal biquad and the oscillator require large capacitors (2X 5nF), difficult to integrate.

Response:

Thank you for the comments. As we mentioned in the abstract “The circuit is designed for low-frequency biomedical and sensor applications” and in introduction is mentioned that our proposed circuit is devoted for low frequency applications like biomedical that attribute a bandwidth from sub hertz up to 10kHz only. The most important for these applications is the low voltage operation capability, low power consumption and the input voltage range without degrading other performance. The 5nF capacitors should be implemented as off chip capacitors. We have added this information to the manuscript.

 A few comments and suggestions for improving the manuscript are provided in the following.

 Abstract - should contain metrics of the performance parameters, tuning range, etc. not only power consumption

Response:

Thank you. More information about the bandwidth of the filter and the THD have been added.

line 177: "therefore the TA can be tuned using the current source Iset". Please provide details regarding the circuit implementation of current sources Iset and IB. Do these current souces have enough voltage headroom: VDD - VGS? What is the VGS of  dide-connected MB transistors?

Response:

Thank you. In the literature, there are several subthreshold voltage current sources, also a digital controlled current source are usually used. However, the CMOS structure of the current source is not the subject of our design. 

The VGS of the diode connected MB is as expected around VDD/2 (155 mV).

Round 2

Reviewer 1 Report

Ok